# Mutations in Both the Surface and Transmembrane Envelope Glycoproteins of the RAV-2 Subgroup B Avian Sarcoma and Leukosis Virus Are Required to Escape the Antiviral Effect of a Secreted Form of the Tvb^S3^ Receptor [note 1]

**DOI:** 10.3390/v11060500

**Published:** 2019-05-31

**Authors:** Xueqian Yin, Deborah C. Melder, William S. Payne, Jerry B. Dodgson, Mark J. Federspiel

**Affiliations:** 1Department of Molecular Medicine, Mayo Clinic, Rochester, MN 55905, USA; Yin.Xueqian@mayo.edu (X.Y.); Melder.Deborah@mayo.edu (D.C.M.); 2Department of Microbiology, Michigan State University, East Lansing, MI 48824, USA; federspiel.mark1@mayo.edu (W.S.P.); dodgson@msu.edu (J.B.D.)

**Keywords:** avian sarcoma and leukosis viruses, receptor usage, envelope glycoprotein evolution

## Abstract

The subgroup A through E avian sarcoma and leukosis viruses ASLV(A) through ASLV(E) are a group of highly related alpharetroviruses that have evolved to use very different host protein families as receptors. We have exploited genetic selection strategies to force the replication-competent ASLVs to naturally evolve and acquire mutations to escape the pressure on virus entry and yield a functional replicating virus. In this study, evolutionary pressure was exerted on ASLV(B) virus entry and replication using a secreted for of its Tvb receptor. As expected, mutations in the ASLV(B) surface glycoprotein hypervariable regions were selected that knocked out the ability for the mutant glycoprotein to bind the sTvb^S3^-IgG inhibitor. However, the subgroup B Rous associated virus 2 (RAV-2) also required additional mutations in the C-terminal end of the SU glycoprotein and multiple regions of TM highlighting the importance of the entire viral envelope glycoprotein trimer structure to mediate the entry process efficiently. These mutations altered the normal two-step ASLV membrane fusion process to enable infection.

## 1. Introduction

The subgroup A through E avian sarcoma and leukosis viruses [ASLV(A) through ASLV(E)] are a group of highly related alpharetroviruses that have evolved their *env* genes encoding the viral envelope glycoproteins from a common ancestor to use members of very different host protein families as receptors to enable efficient virus entry [1,2,3]. All retroviruses initially synthesize their envelope glycoproteins as a precursor that is subsequently processed into two glycoproteins, the surface (SU) and transmembrane (TM) glycoproteins that then form a trimer of SU:TM heterodimers. The SU glycoprotein contains the domains important for interaction with a host protein receptor. The ASLV(A) through ASLV(E) SU glycoproteins are highly conserved except for five hypervariable domains, vr1, vr2, hr1, hr2 and vr3 [4,5,6]. A variety of studies have identified hr1 and hr2 as the principle binding domains between the viral glycoprotein trimer and the host protein receptor, with vr3 contributing to the specificity of the receptor interaction for initiating efficient infection [7,8,9,10,11,12]. The TM glycoprotein contains domains responsible for the fusion process of the viral and cellular membranes necessary for entry and tethers the trimer to the viral surface. Members of three very different families of proteins have been identified to be receptors of these five ASLVs, although all are simple, single-spanning membrane proteins. Tva proteins are related to low density lipoprotein receptors (LDLR) and are receptors for ASLV(A) [13,14,15]. Tvb proteins are related to tumor necrosis factor receptors and are receptors for ASLV(B), ASLV(D) and ASLV(E) [16,17,18,19]. Tvc proteins are related to mammalian butyrophilins, members of the immunoglobulin protein family, and are receptors for ASLV(C) [20,21].

The membrane fusion triggering mechanism of these five ASLV subgroups is a novel combination of the two classic triggering mechanisms of class I enveloped virus fusion proteins [22,23,24,25]. ASLVs require a specific interaction between the viral glycoproteins and receptors at the cell surface at neutral pH to trigger an initial conformational change in the viral glycoproteins, but then require a subsequent expose to low pH to complete the fusion of the viral and cell membranes to effect entry. The triggering of the ASLV viral glycoprotein trimer upon receptor binding results in a conformational change in the SU glycoprotein that presumably separates the SU domains to allow the TM glycoproteins to form an extended structure projecting the fusion peptide (FP) toward the host cell membrane. Two domains in TM, the *N*-terminal heptad repeat (HR1) and the C-terminal heptad repeat (HR2), are critical for the formation of the extended structure. The fusion peptide is thought to interact with the target membrane irreversibly, forming an extended prehairpin TM oligomer structure anchored in both the viral and target membranes. The cooperation of several of these extended TM oligomers is thought to be necessary to complete the fusion process. The viral and target membranes are brought into close proximity when the HR2 repeats fold back into the grooves formed by the HR1 repeats, forming presumably the most stable TM structure, the six-helix bundle (6HB). The fusion of the membranes proceeds through a number of steps and the 6HB may undergo additional structural rearrangements to complete the fusion process.

The sequence of the two-step ASLV viral envelope glycoprotein fusion entry process can be further analyzed using biochemical assays at physiological and non-physiological temperatures [26,27,28,29]. Secreted forms of the ASLV receptors can bind ASLV glycoproteins on virions in vitro at 4oC but require physiological temperatures to trigger the first-step receptor-induced SU conformational changes. The formation of the 6HB TM oligomers then requires the exposure of the receptor-triggered glycoprotein trimers to low pH again at physiological temperatures. Smith et al. [25] formally demonstrated that the mature, ASLV(A) glycoprotein trimer is locked into a metastable conformation that can be triggered to form a 6HB structure at a characteristic high temperature (60–65 °C) a characteristic of metastable conformations, without receptor binding and low pH exposure [25,30]. Therefore, the ASLV envelope glycoproteins not only must evolve to specifically use a receptor with high binding affinity, but also maintain the ability to be productively triggered to allow specific initial structural conformations and then allow further structural conformations to mediate the efficient fusion of the viral and cellular membranes to complete entry.

We have exploited genetic selection strategies to force the replication-competent ASLVs to naturally evolve and acquire mutations to escape the pressure on virus entry and yield a functional replicating virus [8,9,31,32]. In this study, evolutionary pressure was exerted on ASLV(B) virus entry and replication using a Tvb immunoadhesin, sTvb^S3^-mIgG both in cultured cells and in birds as a test of using this immunoadhesin as an antiviral strategy. As expected, mutations in the ASLV(B) SU hypervariable regions were selected that knocked out the ability for the mutant glycoprotein to bind the sTvbS3-IgG inhibitor. However, several unexpected results were also obtained. First, an ASLV with a recombinant subgroup B Env glycoprotein, the SU hypervariable regions of Rous associated virus 2 (RAV-2) fused to the subgroup A Schmidt-Ruppin ASLV (SR-A) C-terminal SU region and TM glycoprotein, was able to productively escape the sTvb^S3^-mIgG entry inhibitor. However, a second ASLV constructed with the entire parental RAV-2 Env glycoproteins could not generate escape mutants even after multiple experiments. Surprisingly, there were major differences in the abilities of the parental RAV-2 glycoprotein trimers compared to the RAV-2/SR-A recombinant Env trimers to be biophysically triggered in vitro to form 6HB TM oligomer structures by heat as well as the normal receptor/low pH exposure two-step fusion process. Further characterization mapped the residues important for these phenotypes to the C-terminal end of the SU glycoprotein and multiple regions of TM highlighting the importance of the entire viral envelope glycoprotein trimer structure to mediate the entry process efficiently.

## 2. Materials and Methods

### 2.1. Soluble Receptor and Retroviral Vector Constructs

The extracellular region of the Tvb^S3^ receptor was amplified by PCR from plasmid pSK101 (provided by John A.T. Young) [16,18] using oligonucleotide primers that converted the ATG start site to an NcoI site (CCATGG), which results in a change of the second amino acid in the leader from Arg to Gly. The PCR fragment was digested with NcoI and EagI and subcloned into the PUCCLA12NCO*stva-mIgG* plasmid replacing the region encoding sTva, creating PUCCLA12NCO*stvb^S3^-mIgG*. The PUCCLA12NCO*stva-mIgG* plasmid, encoding the 83-amino-acid Tva extracellular domain fused to the constant region of the mouse IgG heavy chain was described previously [8]. The mouse IgG domain fused to the sTvb^S3^ domain allowed the use of the extensive anti-mouse IgG reagents to quantitate, purify and track the protein. The soluble receptor gene cassette was isolated as a *Cla*I fragment and cloned into the unique *Cla*I site of the RCASBP(A) retroviral vector. The RCAS family of replication-competent retroviral vectors has been described [33,34]. The *stvb^S3^-mIgG* gene isolated as a *Cla*I fragment was also subcloned into the TFANEO expression vector. TFANEO is a companion expression vector to the RCAS family of retroviral vectors. The expression cassette of TFANEO consists of two LTRs derived from the RCAS vector that provide strong promoter, enhancer, and polyadenylation sites flanking a unique *Cla*I insertion site. The TFANEO plasmid also contains a *neo* resistance gene expressed under the control of the chicken β-actin promoter, and an ampicillin resistance gene for selection in *E. coli*. The RCASBP(A)AP, RCASBP(B)AP and RCASBP(C)AP retroviral vectors which contain the heat stable human placental alkaline phosphatase gene (AP) have been described [33,34].

### 2.2. Cell Culture and Virus Propagation

DF-1 cells [35,36] were grown in DMEM (Gibco/Invitrogen, Walthem, MA, USA) supplemented with 10% fetal bovine serum (Gibco/Invitrogen), 100 units of penicillin per mL, and 100 μg of streptomycin per mL (Quality Biological, Inc, Gaithersburg, MD, USA). In general, DF-1 cultures were passaged 1:3 when confluent. Virus propagation was initiated by calcium phosphate transfection of purified plasmid DNA (10 μg) that contained the retroviral vector in proviral form. Viral spread was monitored by assaying culture supernatants for ASLV capsid (CA) protein by ELISA [37]. Virus stocks were generated from the cell supernatants by clearing the cellular debris by centrifugation at 2000× *g* for 10 min at 4 °C and stored in aliquots at −80 °C. DF-1 cells transfected with the TFANEO plasmid were grown in 500 μg/mL G418 (Gibco/BRL) to select for neomycin-resistant cells. Clones were isolated using cloning cylinders (Bellco Glass Inc., Vineland, NJ, USA), expanded, and maintained with standard medium supplemented with 250 μg/mL G418.

### 2.3. ASLV Alkaline Phosphatase Assays

In a direct AP titer assay, DF-1 cell cultures (~30% confluent) were incubated with 10-fold serial dilutions of the RCASBP/AP virus stocks for 42–48 h at 39 °C. In a pre-absorption AP challenge assay, the 10-fold viral serial dilutions were first mixed with 2 mL of supernatant containing sTvb^S3^-mIgG for 2 hr at 4 °C and then assayed as above. The assay for alkaline phosphatase activity was has been described previously [32].

### 2.4. Immunoprecipitation and Western Transfer Analysis of sTvb^S3^-mIgG Proteins

A 500 μL aliquot of culture supernatant was incubated with 50 μL of anti-mouse IgG-agarose beads (Sigma-Aldrich, St. Louis, MO, USA) for ≥1 h at 4 °C. The sTvb^S3^-mIgG agarose bead complexes were collected by centrifugation and washed twice in dilution buffer [50 mM Tris-buffered saline (TBS), 1% Triton X-100, 1 mg/mL BSA], once in 50 mM TBS and once in 0.05 M Tris Cl, pH 6.8. The washed complexes were collected by centrifugation, resuspended in 50 μL 1× Laemmli buffer (2% SDS, 10% glycerol, 0.05 M Tris Cl, pH 6.8, 0.1% bromophenol blue) without β-mercaptoethanol, and heated for 5 min at 100 °C. The agarose in the samples was collected by centrifugation for 2 min and the supernatants were transferred to new tubes. Prior to gel electrophoresis, 1.0 μL β-mercaptoethanol was added to each 50 μL sample and the samples were heated for 5 min at 100 °C. The denatured immunoprecipitates were separated by 12% SDS-PAGE, and transferred to a nitrocellulose membrane. The filters were blocked with 10% non-fat dry milk (NFDM) in PBS, probed with 0.05 μg/mL peroxidase-conjugated goat anti-mouse IgG antibodies (Kirkegaard and Perry Laboratories, Gaithersburg, MD, USA) in rinse buffer (100 mM NaCl, 10 mM Tris Cl, pH 8, 1 mM EDTA, 0.1% Tween 20) and 1% NFDM, and washed in rinse buffer. Protein/antibody complexes were detected with the Western Blot Chemiluminescence Reagent (Dupont NEN, Boston, MA, USA) according to the manufacturer’s instructions. The immunoblot was then exposed to Kodak X-Omat film.

### 2.5. Mouse IgG ELISA

An ELISA for mouse IgG was described previously [32]. The standard control protein, ImmunoPure^®^ Mouse IgG Fc fragment (Pierce, Rockford, IL, USA), and the sTvb^S3^-mIgG proteins in culture supernatant or chicken serum were serially diluted in blocking buffer. The blocking buffer was removed from the wells and the diluted samples (100 μL/well) added and incubated for 1 hr at 37 °C. The wells were then washed three times with wash buffer and incubated with 0.8 μg/mL goat anti-mouse IgG Fc fragment antibody conjugated to horseradish peroxidase (Pierce) in blocking buffer for 1 hr at 37 °C. The wells were washed three times with wash buffer and incubated with substrate [1.0 mg/mL 5-aminosalicylic acid (Sigma), 18 mM potassium phosphate, monobasic, 2 mM sodium phosphate, dibasic pH 6.0 and 0.01% hydrogen peroxide] for 1 hr at room temperature in subdued light. The absorbance at 490 nm was read with a microtiter plate reader. The linear range for a standard experiment was between 0.5 and 50 ng/mL ImmunoPure^®^ Mouse IgG Fc fragment.

### 2.6. In Vivo ASLV Challenge Assay

Line 0 embryos were somatically infected with RCASBP(A) or RCASBP(A)stvb^S3^-mIgG by injecting unincubated eggs near the blastoderm with 100 μL containing 1 × 10^6^ DF-1 cells producing the virus. Line 0 is a White Leghorn line that is genetically susceptible to all ASLV subgroups except subgroup (E) and is free of endogenous proviruses that are closely related with ASLV. Viremic chicks were identified at hatch by ELISA for the ASLV CA protein. Viremic and uninfected control chicks were infected intra-abdominally with 10^5^ infectious units of either RAV-2 [an ASLV(B) isolate] or RAV-49 [an ASLV(C) isolate]. Blood was collected at 8 weeks post-challenge, and the serum assayed for infectious subgroup (A), (B), or (C) ASLV by the In Vitro ASLV Assay (see below).

### 2.7. *In Vitro* ASLV Assay

The presence of infectious ASLV in chickens was determined by assaying the 8-week serum samples on two different cell lines with different ASLV envelope subgroup susceptibilities: DF-1 cells (C/E) which supports the replication of ASLV(A), ASLV(B), and ASLV(C); and line *alv*6 CEF (C/A) [38] which supports ASLV(B) and ASLV(C) replication. Serum samples (100 μL) were added to the cells and the cells were incubated for 9 days in media (5% serum) to allow ASLV to spread. The media was changed after 3 days to avoid detection of ASLV proteins in the original serum sample. The cells were then solubilized by two cycles of rapid freeze-thaw to release ASLV Gag antigens. The ASLV capsid protein was detected by ELISA. A positive sample was defined as having an optical density reading of >0.200. The In Vitro ASLV Assay can detect infectious ASLV levels as low as 10 IFU/mL.

### 2.8. Cloning and Nucleotide Sequence Analysis of Integrated Viral DNA

DNA was isolated from infected cells in culture using the QIAamp Tissue Kit (Qiagen). The entire *env* gene was amplified by PCR using Taq DNA Polymerase (Promega, Madison, WI, USA) with the primers 5′GGGACGAGGTTATGCCGCTG-3′ (~50 bp upstream of *Asp*718 site) and 5′TACCACCACCCATGTACTGCC-3′ (just downstream of the *env* gene). Each Taq PCR contained 1.25 μL 10× PCR buffer (final concentration, 50 mm Tris-Cl, pH 8.3, 50 mM KCI, 7 mM MgCl_2_ 1.1 mM β-mercaptoethanol), 1.25 μL of 1.7 mg/mL BSA, 0.5 μL of each dNTP at 25 mM, 0.5 μL of each primer (A_260_ = 5), 6.0 μL H_2_0, and 1.0 μL of DNA (genomic DNA ~100 ng/μL; plasmid DNA ~2 ng/μL). The reactions were heated to 90 °C for 1 min and initiated by the addition of 1.5 μL of Taq DNA polymerase diluted 1:10 *v/v* (0.75 units). Thirty cycles of PCR were carried out as follows: 90 °C for 40 s, then 59 °C for 80 s. The amplified products were separated by agarose gel electrophoresis, the ~2.0 kb product purified and cloned into pCR2.1-TOPO using the TOPO TA Cloning kit (Invitrogen).

The nucleotide sequence of the *env* genes was determined by the Mayo Clinic Molecular Biology Core on a Perkin Elmer ABI PRISM 377 DNA sequencer (with XL upgrade) with PE Applied Biosystems ABI PRISM dRhodamine Terminator Cycle Sequencing Ready Reaction Kit and AmpliTaq DNA Polymerase (PE Applied Biosystems, Foster City, CA, USA).

### 2.9. Flourescence-Activated Cell Sorting (FACS) Analysis of Envelope Glycoprotein Binding to Receptor

Uninfected DF-1 cells or DF-1 cells infected with either wild-type or mutant ASLV viruses were removed from culture with Trypsin de Larco (Quality Biological, Inc., Gaithersburg, MD, USA) and washed with Dulbecco’s phosphate buffered saline (PBS). The cells were fixed with 4% paraformaldehyde in PBS at room temperature for 15 min and then washed with PBS. Approximately 1 × 10^6^ cells in PBS supplemented with 1% calf serum (PBS-CS) were incubated with supernatant containing either the sTvb^S3^-mIgG on ice for 30 min. The sTvb^S3^-mIgG protein was produced by the stable DF-1 cell line TF/sTvb^S3^-32. The cells were then washed with PBS-CS and incubated with 5 μL of goat anti-mouse IgG (H + L) linked to phycoerythrin (Kirkegaard and Perry Laboratories) in PBS-CS (1 mL total volume) on ice for 30 min. The cell:soluble receptor-mIgG:Ig-phycoerythrin complexes were washed with PBS-CS, resuspended in 0.5 mL PBS-CS, and analyzed with a FACSCalibur instrument using the CELLQuest 3.1 software (Becton Dickinson, Franklin Lakes, NJ, USA).

### 2.10. Apparent Dissociation constant (K_d_) Calculations

The maximum possible bound fluorescence and K_D_ value for each data set obtained from the FACS binding assays were estimated by fitting the data via non-linear least squares to a log logistic growth curve function: f(y)=M/[1 + e^-r(log*x*-log*Kd*)^] where y is the mean fluorescence, M is the maximum flourescence, r is the rate, x is the concentration of sTva-mIgG and K_d_ is the dissociation constant defined as the concentration of sTva-mIgG at half-maximal binding [8]. The statistical significance among the estimated K_D_ values was analyzed using Analysis of Variance methods. The estimated average K_D_ value for each glycoprotein was obtained along with the associated standard deviation.

### 2.11. SDS-PAGE and Western Immunoblot Analysis

Supernatants from confluent cultures were cleared of cellular debris by centrifugation at 2000× *g* for 10 min at 4 ×. Virions (10 mL of culture supernatant) were pelleted through 1 mL of a 20% sucrose pad (20% sucrose, 100 mM NaCl, 20 mM Tris·Cl, pH 7.5, 1 mM EDTA) by ultracentrifugation at 35,000 rpm in a SW41 rotor (Beckman Coulter, Brea, CA, USA) for 60 min at 4 °C. The viral pellet was resuspended in 100 μL Laemmli loading buffer (2% SDS, 10% glycerol, 50 mM Tris·Cl, pH 6.8, 5% β-mercaptoethanol, 0.1% bromophenol blue) and boiled for 5 min. Viral proteins were separated by 10% SDS-PAGE and transferred to a nitrocellulose membrane. The western transfer filters were blocked in phosphate-buffered saline (PBS) with 10% nonfat dry milk (NFDM) for 1 hr at 25 °C. The filters were then rinsed briefly in rinse buffer (100 mM NaCl, 10 mM Tris Cl, pH 8, 1 mM EDTA, 0.1% Tween 20), and incubated with rabbit anti-ASLV(A) TM peptide sera (1:1000 dilution) in rinse buffer containing 1% NFDM for 1 hr at 25 °C The filters were washed extensively with rinse buffer, and then incubated with 50 ng/mL peroxidase-labeled goat anti-rabbit or goat anti-mouse IgG (H+L) (Kirkegaard and Perry Laboratories, Gaithersburg, MD, USA) in rinse buffer with 1% NFDM for 1 hr at 25 °C. After extensive washing with rinse buffer, immunodetection of the protein-antibody-peroxidase complexes was performed with the Western Blot Chemiluminescence Reagent (DuPont NEN, Boston, MA, USA). The immunoblots were then exposed to Kodak X-Omat film.

### 2.12. TM Oligomerization Assays

The TM oligomerization assays were performed essentially as described by Smith et al. [25]. In short, pelleted virus was resuspended in a small volume of HNC buffer (10 mM HEPES, pH 8.0, 130 mM NaCl, 1 mM CaCl_2_), incubated with or without chicken sTvb^S3^-mIgG (50 nM) 20 min on ice to allow binding. The pH was adjusted with predetermined amounts of 1 M HEPES, pH 7.4 or 4.5 and the samples incubated at 37 °C for 30 min. The samples were then neutralized with 1 M Tris-HCl pH 8.0 or 9.5, lysed with 1% SDS and incubated at 37 °C for 10 min. Samples were then reduced with Laemmli load buffer, separated by SDS-PAGE and analyzed by western immunoblot using an antibody specific for the C-terminus of ASLV(A) TM. The temperature threshold that triggers the TM oligomerization without receptor was determined by incubating the purified virions in HN buffer at the indicated temperature for 30 min or boiled for 5 min (100 °C) prior to lysis in SDS.

## 3. Results and Discussion

### 3.1. The sTvb^S3^-mIgG Receptor Immunoadhesin Protein Can Be Expressed to High Levels and Potently Inhibits ASLV(B) Infection in Cultured Avian Cells

The subgroup A RCASBP(A) retroviral vector that contains and expresses the *stvb^S3^-mIgG* gene used in this study is shown schematically in Figure 1.

Virus propagation was initiated by transfection of the plasmid containing the retroviral vector into DF-1 cells. The transfected cells were passaged when confluent to allow virus to spread throughout the culture resulting in a chronically infected culture that expressed stable levels of the sTvb^S3^-mIgG protein (Table 1). The same *stvb^S3^-mIgG* gene construct was also subcloned into the expression vector TFANEO, which expresses an experimental gene under the control of the ASLV LTRs and contains the *neo* gene under the control of the chicken β-actin promoter (Figure 1A). Clonal DF-1 cell lines were selected that expressed stable levels of sTvb^S3^-mIgG protein. The TF/sTvb^S3^-32 cell line expressed the highest level of sTvb^S3^-mIgG protein and was used in this study (Table 1). We did not observe any cytotoxic effects of expressing high levels of the sTvb^S3^-mIgG receptor protein in DF-1 cells. This was a concern since high levels of RCASBP(B) replication results in a transient period of cytotoxicity not observed with the RCASBP(A) virus.

The sTvb^S3^-mIgG fusion protein has a calculated molecular weight of 41,947 and migrates as a diffuse band (50 to 65 kDa) due to N-linked glycosylation (Figure 1B). The binding affinities of sTvb^S3^-mIgG for subgroup B and D envelope glycoproteins were assayed by FACS using DF-1 cells chronically infected with ASLV(B) or ASLV(D). The sTvb^S3^-mIgG receptor protein bound both subgroup B and D glycoproteins with high affinity (Figure 1C): subgroup B glycoproteins (0.84 ± 0.24 nM) with approximately 3-fold higher affinity compared to subgroup D glycoproteins (2.42 ± 0.55 nM). As expected since Tvb^S3^ is specifically the receptor for subgroup B, D and E ASLVs, the sTvb^S3^-mIgG protein did not bind to envelope glycoproteins expressed on DF-1 cells infected with ASLV(A) or ASLV(C) at experimentally detectable levels. These estimates of sTvb^S3^-mIgG binding affinity may be somewhat high due to the predicted dimer structure of the sTvb^S3^-mIgG immunoadhesin.

The level and specificity of the antiviral effect of the sTvb^S3^-mIgG receptor protein in cultured cells was quantitated by challenging DF-1 cells expressing sTvb^S3^-mIgG from the RCASBP(A) vector, the TF-sTvb^S3^-32 stable cell line expressing the sTvb^S3^-mIgG from a non-viral expression plasmid, DF-1 cells infected with the RCASBP(A) vector alone, and parental DF-1 cells, with the RCASBP(B)AP or RCASBP(C)AP replication-competent viral vectors. Serial dilutions (10-fold) of the viruses were first incubated with supernatants from confluent control or sTvb^S3^-mIgG expressing cultures for 1 h at 4 °C and then plated on the appropriate non-confluent (35% confluent) cells in culture. The infectious titers were determined by assaying for AP activity 2-days after infection. As expected, the sTvb^S3^-mIgG receptor protein specifically inhibited entry of subgroup B ASLV but had no significant effect on subgroup C ASLV entry efficiency (Table 1). In addition, a higher antiviral effect on ASLV(B) entry was observed in the TF/Tvb^S3^-32 cells (725-fold inhibition) that expressed the higher level of the sTvb^S3^-mIgG receptor protein (40 nM) compared to the RCASBP(A)stvb^S3^-mIgG infected cells (70-fold inhibition) that expressed a lower level of sTvb^S3^-mIgG (11 nM).

### 3.2. Delivery, Expression and Antiviral Effect of the sTvb^S3^-mIgG Immunoadhesin on ASLV(B) Challenge in Chickens

We have previously shown that the RCAS vectors can deliver experimental genes to most tissues in the chicken relatively efficiently by infecting embryos in fertile, unincubated eggs by injecting DF-1 cells producing the replication-competent viral vectors near the embryo (~50,000 cell stage) [32]. Chicks viremic for the RCAS vector are then identified after hatch by assaying serum for the ASLV CA protein by ELISA. In this study, two groups of viremic chicks were produced: one group infected with the RCASBP(A) vector alone, and a second group infected with RCASBP(A)stvb^S3^-mIgG and expressing the sTvb^S3^-mIgG receptor protein. There were no observable cytotoxic effects of sTvb^S3^-mIgG receptor protein expression in the embryos, hatched chicks or older chickens: hatch rates and survival of birds infected with RCASBP(A) and RCASBP(A)stvb^S3^-mIgG were comparable.

The RCASBP(A)-alone infected control group was then divided into three groups: five birds were left unchallenged, 15 birds were challenged with 10^5^ ifu RAV-2 (a subgroup B ASLV), and 15 birds challenged with 10^5^ ifu RAV-49 (a subgroup C ASLV). The RCASBP(A)stvb^S3^-mIgG infected group was divided into two groups: 17 birds were challenged with 10^5^ ifu RAV-2, and 22 birds were challenged with 10^5^ ifu RAV-49. Blood was collected from each bird eight weeks after challenge, and the serum assayed for subgroups A, B and C ASLV using the In Vitro ASLV Assay (Table 2).

In the vector alone controls, all of the birds infected with RCASBP(A) alone and still viremic at the end of the experiment, were susceptible to both subgroups B and C ASLV infection and replication. However, viruses were not detected in five birds infected with RCASBP(A) and subsequently challenged with RAV-2 or RAV-49. These birds were technically viremic for ASLV on day 1 after hatch, although expressing relatively low levels of CA compared to the other chicks. We believe that the initial delivery/spread of the virus was not robust enough in these birds to induce immunological tolerance to the ASLV proteins and sustain the viremia once antibody production commenced.

All 22 RCASBP(A)stvb^S3^-mIgG infected birds challenged with RAV-49 were susceptible to this subgroup C ASLV challenge as expected since the sTvb^S3^-mIgG protein is specific for subgroup B ASLVs. Only in the RCASBP(A)stvb^S3^-mIgG infected birds challenged with RAV-2 was an antiviral effect observed: subgroup B ASLV could not be detected in 88% (15/17) of these birds. However, two birds produced detectable RAV-2 in their serum, 6590 and 6620, perhaps coincidentally they were two of the three birds in the group that expressed the lowest levels of sTvb^S3^-mIgG in their post-hatch serum samples.

### 3.3. Analysis of the Subgroup B Virus Populations in the Serum of RAV-2 Challenged Birds

Why were birds 6590 and 6620 not completely protected from RAV-2 challenge? Was this the result of incomplete sTvb^S3^-mIgG gene delivery and/or low expression? Or, do these birds now have mutant RAV-2 virus populations that have escaped the antiviral effect of sTvb^S3^-mIgG? To answer these questions, we set out to analyze the ASLV(B) populations in the serum of the two birds viremic for ASLV(B), 6590 and 6620, as well as a representative control bird infected with RCASBP(A) vector alone, 6461.

We analyzed the ASLV virus populations in the serum of the birds with both a subgroup A virus, the RCASBP(A) delivery vector, and a subgroup B virus, the RAV-2 challenge virus, by first preferentially amplified subgroup B ASLVs by first passaging the serum in TF/SU(A)-19 cells, a DF-1 cell line expressing SUA-rIgG that is >180,000-fold resistant to ASLV(A) infection. Specifically, ASLVs in 8-week serum samples (100 uL) amplified for three passages (2-days per passage at 1:3 splits) on DF-1 and TF/SU(A)-19 cells [31]: both subgroup A and B viruses should replicate in DF-1 cells, while subgroup B viruses should preferentially replicate in TF/SU(A)-19 cells. As expected all three birds contained ASLVs that rapidly replicated in DF-1 cells, while only birds 6461, 6590 and 6620 contained virus that replicated well in subgroup A ASLV resistant cells although most likely still contain low levels of subgroup A virus. To test whether mutant subgroup B ASLVs resistant to the antiviral effect of sTvb^S3^-mIgG had been selected in these birds, viruses from TF/SU(A)-19 culture supernatants (day 6) were again propagated for three passages in TF/sTvb^S3^-32 cells. Only viruses from bird 6620 clearly showed resistance to the antiviral effect of sTvb^S3^-mIgG early in the infection (days 2 and 4); however, virus from the other two birds did replicate in these cells, most likely due to residual subgroup A virus. The ASLV *env* genes were amplified by PCR from genomic DNA isolated from day 6 infected TF/SU(A)-19 cells, cloned, and the nucleotide sequences of the ASLV *env* genes determined. Both subgroup A and subgroup B *env* genes were amplified. The complete nucleotide sequence was then determined from 7-10 subgroup B *env* gene clones from each bird.

Since only an incomplete nucleotide sequence encoding the RAV-2 SU envelope glycoprotein had been published previously, we initially analyzed subgroup B *env* clones from bird 6461 to establish a complete ‘parental’ RAV-2 *env* gene nucleotide sequence. We analyzed seven independent clones (Figure 2): 4/7 clones contained a RAV-2 *env* gene; 3/7 clones were obvious recombinants between the *env* of RAV-2 and the subgroup A *env* of RCASBP(A). The 6461 RAV-2 nucleotide sequence contained 2-3 nucleotide differences compared to the published RAV-2 sequence.

Two of these nucleotide sequence differences were silent changes; however, approximately one-half of the 6461 clones (4/7) and the published RAV-2 sequence coded for arginine (AGA) at amino acid 70 while 3/7 6461 clones coded for lysine (AAA). The amino acid sequence predicted from the 6461 RAV-2 *env* nucleotide sequence with R70 is shown in Figure 3 as previously published for RAV-2 (red asterisk). The recombinant RAV-2 and RCASBP(A) *env* genes were essentially RAV-2 *env* genes with ~300 nucleotides of the subgroup A *env* gene downstream of the hypervariable regions of SU but upstream of the fusion peptide region of TM (see Figure 2), a region highly conserved between the different ASLV subgroups. However, there are 6 amino acid differences between RAV-2 and RCASBP(A) in this region (Figure 3). It is interesting to note that the RCASBP(B) vector includes this region of RCASBP(A) plus the rest of the SR-A TM.

The subgroup B *env* genes isolated from bird 6590, infected with RCASBP(A)stvb^S3^-mIgG and producing detectable RAV-2 after challenge, were a mixture of “wild-type” RAV-2 genes and similar RAV-2/RCASBP(A) recombinants seen in bird 6461 (Figure 2). In this bird, all of the “wild-type” RAV-2 genes encoded lysine at position 70 (K70). No other mutations were detected so we concluded that in bird 6590, the delivery and/or expression of the sTvb^S3^-mIgG inhibitor was not complete and allowed RAV-2 infection and spread.

### 3.4. Mutations in Several Different Regions in RAV-2 Env Were Required for Resistance to the sTvb^S3^-mIgG Antiviral Effect

To test whether any of the subgroup B *env* genes amplified from bird 6620 coded for envelope glycoproteins resistant to the antiviral effect of sTvb^S3^-mIgG, RCASBP/AP vectors were constructed with the wild-type RAV-2 *env* gene, the 6620-17 *env* gene that contains R196K, Y201N, 27 bp duplication in hr2 and the SU carboxy-terminus region of subgroup A *env*, the 6620-16 *env* gene that contains only the SU carboxy-terminus region of subgroup A *env*, and the 6620-18 *env* gene that contains only the 27 bp duplication in hr2. TF/sTvb^S3^-32 cells were transfected with the plasmids encoding viruses with the wild-type RAV-2 *env* or clone 16, 17 or 18 mutant *env* and the cells passaged to allow virus replication and spread. Only TF/sTvb^S3^-32 cells transfected with clone 6620-17 supported virus replication but only at very low levels until after ~15-days when virus replication rapidly increased.

This implied that the 6620-17 virus was at least partially resistant to sTvb^S3^-mIgG inhibitor, especially in the 6620 bird that expressed much lower levels compared to the TF/sTvb^S3^-32 cells, but the virus may need additional mutations to efficiently escape the high sTvb^S3^-mIgG levels in the TF/sTvb^S3^-32 cells.

TF/sTvb^S3^-32 cells were transfected with the plasmids encoding viruses with the wild-type RAV-2 *env*, 6620-17 env or RCASBP(A) as a positive control. The transfected cells were passaged when confluent to allow virus replication and spread. As expected since the antiviral effect of sTvb^S3^-mIgG is specific for subgroups B, D and E ASLVs, the RCASBP(A) control rapidly established a chronically infected culture of TF/sTvb^S3^-32 cells. Only the 6620-17 mutant virus but not the parental RAV-2 virus was able to replicate in TF/sTvb^S3^-32 cells in the 5-week experiment (Figure 4A). Only low levels of the 6620-17 virus were detected until day 25 post-transfection when a significant increase in virus production was observed. The significant increase in the production of 6620-17 virus after 3-weeks of replication in TF/sTvb^S3^-32 cells implies that the virus may have evolved to increase its resistance to the high concentration of sTvb^S3^-mIgG by acquiring additional mutations in the *env* gene. To test if an escape mutant virus population was selected, day 35 cell culture supernatant (100 ul) from 6620-17 infected TF/sTvb^S3^-32 cells was used to infect fresh TF/sTvb^S3^-32 cells, and the infected cells passaged when confluent to allow virus replication and spread (Figure 4B). This 6620-17 virus population rapidly replicated in fresh TF/sTvb^S3^-32 cells without a lag period further implying that additional mutations were acquired that improve resistance to sTvb^S3^-mIgG.

The ASLV *env* genes were amplified by PCR from genomic DNA isolated from day 35 TF/sTvb^S3^-32 cells transfected with 6620-17 (Figure 4A) and day 11 TF/sTvb^S3^-32 cells infected with day 35 6620-17 supernatant (Figure 4B), and cloned. The nucleotide sequences of the ASLV *env* genes were determined from at least ten individual clones from each culture, and the corresponding amino acid sequences compared to the original 6620-17 mutant protein encoded by the plasmid used to initiate infection. The virus population produced with the original 6620-17 Env sequence did evolve after prolonged initial passage in TF/sTvbS3-32 cells to a population dominated (10/11 clones) with the 6620-17 Env but with two new, additional mutations: E266K mutation in the SU vr3 variable region, and the G441R mutation in the critical chain reversal region between the two heptad repeats HR1 and HR2 in the TM glycoprotein.

To characterize these new mutations, three new RCASBP plasmids were constructed containing the 6620-17 *env* gene adding the E266K mutation (EK), or 6620-17 adding the G441R (GR) mutation or adding both mutations (EKGR) in the 6620-17 Env background. These three plasmids plus the control RAV-2 plasmid were transfected into normal DF-1 cells to assess viral fitness (Figure 4C). Both 6620-17 + EKGR and 6620-17 + g and the control RAV-2 viruses replicated well in DF-1 cells without significant time lags in reaching maximum virus levels, with no additional genomic changes were found in these viruses. The 6620-17 parent virus replicated with a slight delay in reaching maximum levels and the genomes of 10/10 clones had all lost the 9 amino acid duplication in SU hr2 region leading to the possible conclusion that without the sTvb^S3^-IgG inhibitor, the 9 AA duplication is not needed and its deletion results in a more fit virus. Finally, the replication of the 6620-17+EK virus required a significant delay before virus replication reach high levels. The resulting virus pool had several genotypes with the majority of genomes (7/10) acquiring two mutations: the D320G mutation in the C-terminal region of SU and the A423V mutation in the HR1 region. The E266K mutation knocks out the ability of the 6620-17 + EK and the 6620-17 + EKGR Env glycoproteins to detectably bind to the sTvb^S3^ receptor while 6620-17+GR and RAV-2 bind the sTvb^S3^ receptor with wild-type affinities.

The 6620-17, 6620-17+EK, 6620-17+GR and 6620-17 + EKGR plasmids plus the control RAV-2 plasmid were also transfected into TF/sTvbS3-32 cells to assess viral resistance to the sTvb^S3^-IgG inhibitor (Figure 4D). Only the 6620-17 + EKGR virus containing both the E266K mutation that knocks out the ability of the sTvb^S3^-IgG inhibitor to bind the ASLV(B) Env, and the G441R mutation in the chain reversal region of the TM, replicated efficiently in the presence of the sTvb^S3^-IgG in TF/sTvb^S3^-32 cells. Both of the 6620-17 viruses with single mutations required a significant time delay before either virus replicated efficiently with 10/10 clones of the 6620-17 + GR virus acquiring the E266K mutation, and the 6620-17 + EK virus acquiring one or more mutations in TM, A423V and or G441E. Again, as observed in the experiment of Figure 4A, a detectable level of the wild-type RAV-2 virus was again never produced after transfection of TF/sTvbS3-32 cells.

### 3.5. Residues in the C-Terminal End of SU and in TM in RAV-2 Alter the Biophysical Properties of Virus Entry

The subgroup B RCASBP(B) vector was originally constructed by replacing just the N-terminal region of SR-A RCASBP(A) Env with the complementary region of RAV-2 SU. At the time, only the nucleotide sequence of this region of RAV-2 Env had been determined and cloned, and this ASLV Env region of SU had been shown to confer receptor specificity. Therefore, RCASBP(B) Env is composed of the first ~280 residues of RAV-2 SU fused to the SR-A Env (see Figure 3 for comparison to the complete RAV-2 Env determined in this study). The extracellular amino acid sequence is highly conserved in the C-terminal of the ASLV SU hypervariable regions (Figure 3), although there are 8 extracellular amino acid differences between RAV-2 and SR-A in this region. The region acquired by the 6620-17 virus from the SR-A RCASBP(A) vector contained 6/8 extracellular amino acid Env differences between SR-A and RAV-2 (Figure 3).

Since both viruses replicate at approximately the same rate and to the same titers in DF-1 cells, we hypothesized that the amino acid differences between RAV-2 and RCASBP(B) altered the viruses fundamental ability to promote virus entry in some way, likely altering the two-step mechanism of ASLV Env fusion process for entry. The conformation of the mature, metastable RAV-2 and RCASBP(B) glycoprotein trimers on wild-type virions purified from DF-1 cells were analyzed using the TM oligomerization assay conditions based on experimental conditions defined by Smith et al. [25]. The heat denaturation of RCASBP(B) glycoprotein trimers triggered the trimers to form stable, SDS-resistant TM oligomers initially at ~55 °C, but predominately at 60–65 °C, with almost all TM in oligomers at 65 °C: TM monomers (~30 kDa) form two oligomeric species of 70–80 kDa and ~170 kDa (Figure 5A). The heat denaturation of RAV-2 glycoprotein trimers also formed TM oligomers at predominately 60–65 °C, but a low but significant level of the 70–80 kDa TM oligomer was evident at all temperatures tested, as low as 37 °C, indicating the RAV-2 glycoprotein trimer structure was not as stable compared to the RCASBP(B) glycoprotein trimers.

The combination of receptor-induced conformation changes and a subsequent low pH exposure of RCASBP(B) envelope glycoprotein trimers at 37 °C produced TM oligomers as expected: the combination of first receptor binding at 37 °C to triggering structural changes in SU, followed by low pH exposure (pH 5.0) efficiently produced the highest levels of TM oligomers (Figure 5B). In contrast, the RAV-2 glycoprotein trimers again did not demonstrate the same level of specificity as RCASBP(B) glycoprotein trimers for receptor binding followed by low pH exposure to efficiently produce stable TM oligomers. Again, a low level of TM oligomers was observed at 37 °C without receptor binding and at neutral pH 7.4, and pH 5.0 exposure did trigger a higher level of TM oligomers without receptor binding. However, RAV-2 glycoprotein trimers upon binding receptor do not appear to require a low pH exposure to produce maximum levels of stable TM oligomers since exposure to pH 7.4 and pH 5.0 resulted in similar TM oligomer levels.

To determine if the 8-extracellular amino acid differences between RCASBP(B) and RAV-2 (see Figure 3) were responsible for their different biophysical properties, the SR-A Env glycoprotein residues of RCASBP(B) were converted to RAV-2 residues generating RCASBP(B) + 8, and the reverse for RAV-2 generating RAV-2 + 8. DF-1 cells were transfected with these plasmids, cultured, and virions harvested and purified and the properties of their metastable glycoprotein trimers analyzed using heat denaturation TM oligomerization assays. Changing these 8-residues reversed the characteristic heat denaturation pattern of TM oligomer formation to the virus providing the 8-amino acid residues: the assay result of RCASBP(B) + 8 was now identical to RAV-2; while the assay result of RAV-2 + 8 was identical to RCASBP(B) (Figure 5A). Since four of the eight amino acid differences are conservative amino acid substitutions, two new viruses were constructed that just changed the four non-conserved differences to generate RCASBP(B) + 4 and RAV-2 + 4 viruses. Similar results were obtained with just these four non-conserved amino acid changes: the heat denaturation TM oligomer assay result of RCASBP(B)+4 was now identical to RAV-2; while the assay result of RAV-2 + 4 was identical to RCASBP(B). The combination receptor-induced conformation changes and a subsequent low pH exposure of purified RCASBP(B)+4 and RAV-2 + 4 viruses at 37 °C also reversed the assay results where RCASBP(B) + 4 was now identical to RAV-2; while the assay result of RAV-2 + 4 was identical to RCASBP(B) (data not shown).

These data imply these four amino acid differences produce structural changes in these regions of the SU and TM glycoproteins that alter the interplay between SU and TM glycoproteins in the Env trimer that alter the efficiency of virus entry and ability to adapt/evolve their receptor usage to escape the sTvb^S3^-mIgG inhibitor.

### 3.6. The 136-142 hr1 Deletion in RCASBP(B) Env That Rescued the RCASBP(B) Virus from the sTvb^S3^-IgG Inhibitor Does Not Rescue RAV-2

To characterize the possible effects of amino acid differences in this region on the ability of ASLV to escape the antiviral effect of sTvb^S3^-IgG, TF/sTvbS3-32 cells were transfected with RAV-2 (RCASBP containing the entire RAV-2 *env* gene) and RCASBP(B) plasmids and passaged to select for possible escape variants. After a lag period of ~15-days, rapidly replicating virus was detected in the RCASBP(B) transfected culture (Figure 6A). However, as was observed in the previous two experiments, no virus was produced in the cells transfected with RAV-2. The RCASBP(B) *env* genes cloned from the apparent escape virus pool all contained the same deletion, deleting residues 136-142 in the hr1 region of SU. Both RAV-2 and RCASBP(B) contain this same SU variable regions but only viable escape mutations occurred in the RCASBP(B) virus background. An RCASBP(B) virus was constructed with the 136-142 hr1 deletion, RCASBP(B)/Del136-142: this deletion knocks out the ability of the RCASBP(B)/Del136-142 Env glycoproteins to detectably bind to the sTvb^S3^ receptor while RCASBP(B) Env glycoproteins bind the sTvb^S3^ receptor with wild-type affinity.

Since multiple attempts have failed to rescue a RAV-2 based virus in the presence of the sTvb^S3^-IgG inhibitor using TF/sTvbS3-32 cells, we constructed RAV-2 with this same hr1 136-142 deletion, RAV-2/Del136-142. The four plasmids encoding the wild-type and deletion mutant viruses were transfected into DF-1 cells and TF/sTvbS3-32 cells, passaged, and the subsequent viruses produced analyzed (Figure 6B). RCASBP(B) and RCASBP(B)/Del136-142 replicated well in DF-1 cells without production lags and without new Env mutations being selected. As expected, RCASBP(B)/Del136-142 replicated well in the presence of the sTvb^S3^-IgG inhibitor in TF/sTvbS3-32 cells without additional Env mutations, while a lag in the production of virus was observed in parent RCASBP(B) culture with selection of the same 136-142 deletion in the escaped viruses as the previous experiment.

The parental RAV-2 virus replicated well in DF-1 cells without genomic changes, and as in previous experiments, RAV-2 did not replicate in the presence of the sTvb^S3^-IgG inhibitor in TF/sTvbS3-32 cells, and no escape RAV-2 virus variants were selected (Figure 6B). Unexpectedly, the constructed RAV-2/Del136-142 virus did not replicate well even in DF-1, with virus production lagging for ~15-days before a significant replication improvement indicating a selection of possible new glycoprotein mutations. The RAV-2/Del136-142 virus pool from DF-1 cells contained a mixture of the parental RAV-2/Del136-142 virus, and viruses with two additional mutations, V359M located in the N-terminal end of the TM glycoprotein, and A382T located in the fusion peptide (Figure 3). In the presence of the sTvb^S3^-IgG inhibitor in TF/sTvbS3-32 cells, the hr1 deletion in the constructed RAV-2/Del136-142 did not directly rescue virus propagation, but a pool of escape virus was selected that retained the Del136-142 deletion and predominately added four mutations: S337L in the C-terminal end of SU, the same V359M and A382T TM mutations, and an additional mutation S413N located in the heptad repeat 1 (HR1) region of TM. This was the first instance that an RAV-2 based virus escaped the antiviral effects of the sTvb^S3^-IgG inhibitor but required the hr1 deletion as a starting point for evolution.

### 3.7. The Selected RAV-2/Del136-142 Escape Mutants Have Altered Receptor and Host Range Specificities

To determine if all the selected mutations are required for efficient RAV-2/Del136-142 virus replication, the plasmid encoding RAV-2/Del136-142 virus was engineered with all four selected mutations, S337L + VAS, the three mutations in TM, +VAS, and only the SU mutation S337L, and the four plasmids transfected into DF-1 and TF/sTvbS3-32 cells (Figure 7A). Both the RAV-2/Del136-142 virus with all four mutations +S337L + VAS and only the three TM mutations + VAS, replicated well in both DF-1 and TF/sTvbS3-32 cells. The SU S337L mutation when present alone in RAV-2/Del136-142 does not improve replication in either DF-1 or TF/sTvbS3-32 cells.

The titers of the RAV-2, RAV-2/Del+VAS and RAV-2/Del+SVAS viruses produced by the transfected DF-1 cells that did not appear to lag in replication (see Figure 7A) were determined using a receptor interference assay using uninfected DF-1 cells and DF-1 cell cultures previously infected with RCASBP(A), RCASBP(B), RCASBP(C) or a subgroup J ASLV. The RAV-2 and RCASBP(B) controls demonstrate the expected receptor interference assay result of a subgroup B ASLV only being blocked by DF-1 cells previously infected by another subgroup B ASLV (Figure 7B).

However, both the RAV-2/Del+VAS and RAV-2/Del+SVAS viruses can infect DF-1/B cells as efficiently as uninfected DF-1 cells, and both viruses are interfered with the infection of DF-1/C cells where the Tvc receptor would be blocked, producing ~20-fold lower titers compared to the other cells. A panel of mammalian cell lines and DF-1 cells were used to determine if the host range of the mutant viruses had also been altered. The RAV-2 control demonstrates the usual ASLV(B) host range restriction of ASLV to avian cells since mammalian cells do not express ASLV receptors, a 4-log reduction in infection efficiency (Figure 7C). However, the +VAS mutations in the TM glycoprotein enable both the RAV-2/Del + VAS and RAV-2/Del + SVAS viruses to infect several mammalian cell lines at levels >10-fold higher compared to the viruses ability to infect DF-1 cells.

### 3.8. The Selected Mutations in the RAV-2/Del136-142 TM Glycoprotein Alter the Biophysical Properties of the RAV-2/Del136-142 Env Glycoprotein Trimer to Allow Efficient Entry, Evasion of the sTvb^S3^-IgG Inhibitor, and Productive Virus Replication

To determine the minimum mutations required for rescue of RAV-2/Del136-142 virus replication, viruses were generated with combinations of single and double mutations of the three VAS TM mutations, and the plasmids transfected into DF-1 and TF/sTvbS3-32 cells (Figure 8A). Only the RAV-2/Del + VA and RAV-2/Del + A viruses replicated relatively well in both DF-1 and TF/sTvbS3-32 cells without significant lags in virus production. The V359M and A382T mutations appear to be the minimum changes required for recovering relatively efficient RAV-2/Del136-142 virus replication.

We hypothesized that the escape mutations selected in TM altered the metastable and triggering properties of the RAV-2/Del136-142 glycoprotein trimers to provide an advantage to increase the efficiency of viral entry and subsequent propagation. The same mutations were also constructed in the RAV-2 virus background to separate the possible effects from the Del136-142 mutation. To characterize the biophysical properties of the viral glycoprotein trimers, TM oligomerization assays were done with virions purified from 2-day transient transfections to ensure that the predominant virus species would be encoded by the plasmid and before any possible selection of additional mutations (Figure 8B). In general, we observe a slight difference in the results of the temperature denaturation TM oligomerization assays when using virus purified from transient transfections: slightly more ‘background’ TM oligomers are observed at lower temperatures; and a relative increase in the TM oligomer levels at 55 °C for RCASBP(B) (compare RACSBP(B) in Figure 8B to Figure 5A). The temperature denaturation of RCASBP(B) and RCASBP(B)/Del136-142 produced a similar pattern of TM oligomers as would be expected since the Del136-142 likely predominately only alters sTvb^S3^-IgG receptor binding but not stability of the fundamental trimer structure. For RAV-2 virions purified from transient transfections, similar levels of TM oligomers are produced at all temperatures (compare RAV-2 in Figure 8B to Figure 5A). For RAV-2/Del136-142, the trimer does appear to form higher levels of TM oligomers at all temperatures compared to RAV-2 indicating the Del136-142 deletion does alter the stability of the trimer and may explain the lack of initial replication in DF-1 cells.

As hypothesized, the mutations acquired in the TM glycoprotein of RAV-2/Del136-142 altered the stability and triggering properties of the both the RAV-2 parental and the RAV-2/Del136-142 Env glycoprotein trimers. The single V359M mutation (+V) and the single A382T (+A) mutation altered the heat denaturation TM oligomer assay results of both the RAV-2/Del136-142 and RAV-2 Env trimers to resemble the RCASBP(B)/Del136-142 and RCASBP(B) Env trimer results upon heat denaturation although RAV-2 + A appears to produce lower levels of TM oligomers compared to RAV-2 + V (Figure 8B). However, both RAV-2/Del136-142 and RAV-2 Env trimers that contain both the V350M and A382T mutations (+VA) do not form high levels of TM oligomers at any temperature even up to 90 °C. RAV-2/Del136-142 and RAV-2 Env trimers that contain the three TM mutations (+VAS) and all four mutations (+SVAS) also do not produce significant levels of TM oligomers even up to 90 °C and look similar to the +VA results.

The TM mutations also altered the RAV-2 and RAV-2/Del136-142 responses to Env trimer triggering by receptor binding and exposure to low pH (Figure 8C). The single V359M mutation altered RAV-2 Env trimer TM oligomer formation to specifically require sTvb receptor binding and pH 5.0 exposure to optimally trigger TM oligomer formation in a pattern very similar to RCASBP(B) (compare RAV-2 + V Figure 8C to RCASBP(B) Figure 5B). As with the V359M mutation, the single A382T mutation also altered RAV-2 Env trimer TM oligomer formation to require both receptor binding and low pH exposure (RAV-2 + A), but the overall levels of TM oligomer formed were much lower compared to RAV-2+V and RCASBP(B). The V359M mutation and A382T single mutations in the RAV-2/Del136-142 background also altered the receptor pH TM oligomer formation, but in this case the RAV-2/Del+V Env trimer and RAV-2/Del + A was only triggered to form TM oligomers by exposure to pH 5.0 without first requiring receptor binding and SU structural changes. As was observed in the heat denaturation TM oligomer assays, both RAV-2/Del136-142 and RAV-2 Env trimers that contain both the V350M and A382T mutations (+VA) do not form high levels of TM oligomers under any experimental condition of receptor binding and low pH exposure. RAV-2/Del136-142 and RAV-2 Env trimers that contain the three TM mutations (+VAS) and all four mutations (+SVAS) also do not appear to produce high levels of TM oligomers and look similar to the +VA results.

## 4. Conclusions

Historically, the RCAS vectors were constructed using the SR-A virus and consequently have subgroup A ASLV envelope glycoproteins, RCAS(A) [33,34]. RCAS vectors with other envelope glycoprotein subgroups were constructed by replacing the *env* gene segment containing the RCAS(A) SU hypervariable regions with the corresponding segment of another ASLV subgroup; the subgroup B fragment from RAV-2 was used to create RCAS(B). It was known at the time that the SU hypervariable region was sufficient to determine receptor specificity. In addition, the C-terminal region of SU glycoprotein and the extracellular region of the TM glycoprotein were nearly identical across the subgroup A-E ASLVs. However, in this study, we surprisingly found a major difference in the ability of RCAS containing the parental RAV-2 *env* gene compared to the RCAS with the recombinant RAV-2/SR-A *env* gene to evolve/adapt their envelope glycoproteins to evade the high affinity binding of a soluble Tvb receptor mimic, the sTvb^S3^-mIgG immunoadhesin, to efficiently infect and replicate in chicken cells.

As expected from other related genetic selections of ASLVs, mutations were selected in the RAV-2 SU glycoprotein hypervariable regions that altered the binding affinity between RAV-2 and the subgroup B receptor sTvb^S3^-mIgG inhibitor. The results of this study also highlight the important interplay of all regions of the SU and TM glycoproteins in the metastable trimer to efficiently promote entry with the minor differences in RAV-2 C-terminal SU and TM glycoproteins significantly altering the biochemical properties of fusion process compared to another ASLV region. Similar mutations in this region of the TM glycoprotein were observed in an ASLV(C) virus with expanded receptor usage that altered TM oligomer formation to form with only low pH exposure and lower temperatures, with no requirement for an initial receptor-induced conformational change [39]. These mutations seemed to make the glycoprotein trimer less stable and more easily triggered to form TM oligomers. Our identified mutations in RAV-2 TM glycoprotein also enable triggering to TM oligomers without receptor interaction and only low pH exposure but appear to be structurally more stable requiring higher temperatures to trigger conformation changes. This apparent difference may be due to the different biochemical conformation assays used by the two labs since the mutant viruses both were in a pre-triggered form that allowed only low pH exposure to advance the fusion process.

It is interesting that the subgroup A-E ASLVs have evolved to specifically use a single receptor with very high affinity rather than have a broader receptor usage. In these genetic selection experiments, we and others have often noted the significant increase in virus-induced cytotoxicity in cultured avian cells with ASLVs that have been selected with a broadened receptor usage [9,40]. This indicates that a broad use of multiple cellular proteins as receptors is a disadvantage and therefore it is likely an advantage for the virus to specifically interact with only one host protein since this interaction may alter the normal abundance/functions of that host protein. Since small changes to the ASLV glycoproteins allow the use of very different proteins as receptors, there may be some unknown structural motifs shared between the receptors that would allow low efficiency use of each receptor family by other subgroup viruses. Indeed, even with the wild-type subgroup B ASLV receptor interference patterns, cells pre-infected with subgroup C virus already interferes 5-10-fold (Figure 7B).

Finally, the subgroup A-E ASLVs and the many highly related variants offer an expansive experimental system to study the evolution of the interactions between viral glycoproteins and host cell proteins used as receptors. The two-step fusion process of these ASLVs, a required specific interaction between the viral glycoproteins and host receptors at the cell surface at neutral pH to trigger an initial conformational change in the viral glycoproteins, followed by a required subsequent expose to low pH to complete the fusion of the viral and cell membranes to effect entry, offers additional windows into the steps of virus entry.

## Figures and Tables

**Figure 1 viruses-11-00500-f001:**
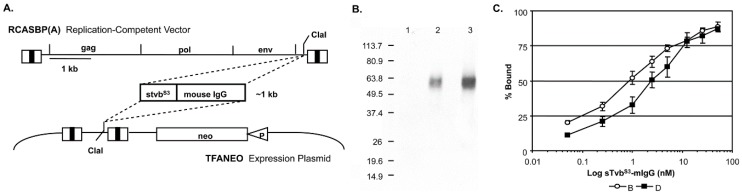
Delivery, expression and function of the sTvb^S3^-mIgG protein. (**A**) Schematic representation of the gene encoding the sTvb^S3^-mIgG protein and the RCASBP(A) viral delivery vector and the TFANEO protein expression construct. (**B**) Western immunoblot analysis of the sTvb^S3^-mIgG protein expression levels after immunoprecipitation using anti-mouse IgG agarose beads and separation using 12% SDS-PAGE from DF-1 culture supernatants. The filter was probed with HRP-conjugated goat anti-mouse IgG, and the bound antibody:protein complexes visualized by chemiluminescence on Kodak X-Omat film. Lane 1: parental DF-1 cells; Lane 2: DF-1 cells infected with RCASBP(A)-sTvb^S3^-mIgG; Lane 3: TF/sTvb^S3^-32 stable DF-1 cell line expressing TFANEO-sTvb^S3^-mIgG. Molecular weight markers in kilodaltons. (**C**) Binding of sTvb^S3^-mIgG to ASLV(B) and ASLV(D) envelope glycoproteins on the surface of DF-1 cells infected with RCASBP(B) or RCASBP(D). The cells were fixed with paraformaldehyde, incubated with different amounts of the sTvb^S3^-mIgG protein, and the envelope glycoprotein:sTvb^S3^-mIgG complexes were bound to an anti-mouse Ig conjugated to phycoerythrin. The levels of phycoerythrin bound to washed cells was measured by FACS. The data are plotted as percent maximum fluorescence bound against sTvb^S3^-mIgG concentration. The values shown are the average of three different experiments with the error bars showing standard error.

**Figure 2 viruses-11-00500-f002:**
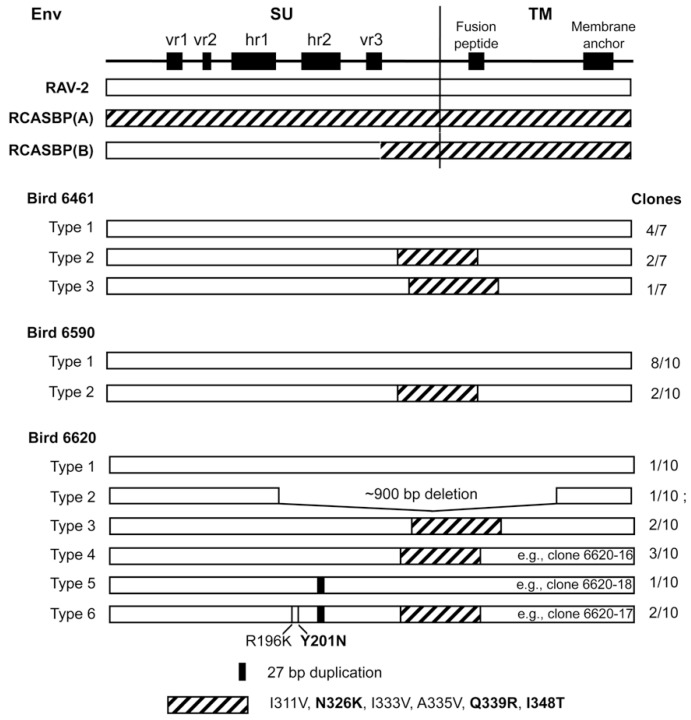
Schematic representations of the envelope glycoproteins (Env) of the parental ASLVs and the cloned ASLV recovered from RAV-2 challenged birds. Bird 6461 was infected with RCASBP(A) as an embryo and then challenged with RAV-2 after hatch. Birds 6590 and 6620 were infected as embryos with RCASBP(A)-sTvb^S3^-mIgG and then challenged with RAV-2 after hatch. Serums from these three birds were used to infect avian cells to amplify and select for subgroup B viruses, the *env* genes PCR amplified from DNA isolated from the infected cells, cloned, and the nucleotide sequences determined from 10 clones from each bird.

**Figure 3 viruses-11-00500-f003:**
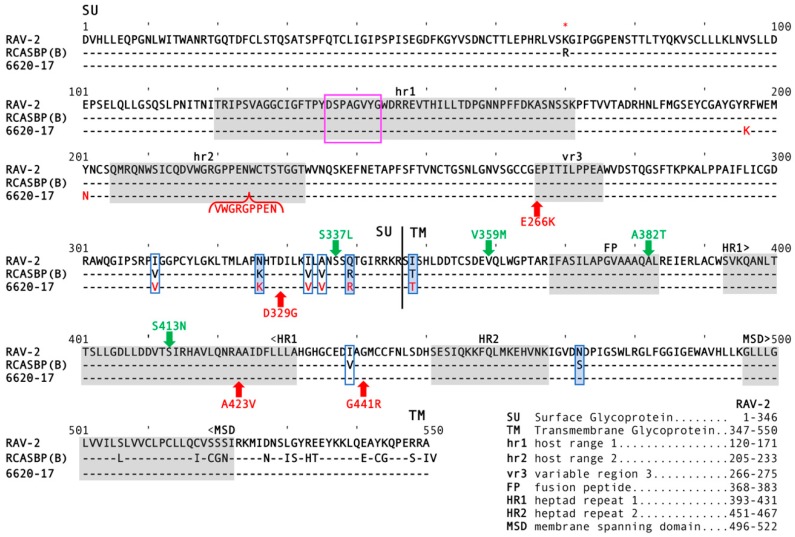
Comparisons of envelope glycoprotein sequences of ASLV(B) viruses compared to RAV-2 sequence. Only differences in the protein sequences are listed. The RAV-2 parental sequence was determined from the sequences of Bird 6461 *env* clones (Figure 2): at residue 70 there was an equal distribution of K70 and R70 (marked with red asterisk). The hypervariable regions of SU glycoprotein (vr1, vr2, hr1, hr2 and vr3) and the functional regions of the TM glycoprotein (FP, HR1, HR2 and MSD) are shaded gray. RCASBP(B) is a combination of RAV-2 sequence (residues 1–279) and RCASBP(A) sequence (residues 280–551). The eight extracellular residue difference between RAV-2 and RCASBP(B) are highlighted with blue boxes: conserved changes (open blue boxes), non-conserved changes (shaded blue boxes). The mutations in 6620-17 are highlighted in red including the 9-amino acid insertion at position 224; the additional mutations selected after addition passage are in red and marked with red block arrows. The 7-amino acid deletion in hr1 (residues 136–142) in RCASBP(B) selected to escape sTvb^S3^-mIgG inhibition by is highlighted in a magenta box. The mutations selected in RAV-2/Del136-142 to improve virus replication are in green and marked with green block arrows. The cloned subgroup B *env* genes amplified from bird 6620 were more diverse compared to bird 6461, with the majority of clones (7 of 10) recombinant RAV-2/RCASBP(A) *env* genes. There was only 1 clone out of 10 that contained a “wild-type” RAV-2 *env* gene. Three clones contained a 27-bp duplication in the SU hr2 hypervariable region resulting in a 9-amino acid duplication between residues 223/224 in RAV-2 Env. Two of the 3 clones with the duplication also contained 2 nucleotide point mutations resulting in R196K and Y201N mutations in the region between the SU hr1 and hr2 hypervariable regions, and were also RAV-2/RCASBP(A) recombinants: the amino acid sequence predicted from the *env* nucleotide sequence of one of these clones, 6620-17, is compared to RAV-2 and RCASBP(B) in Figure 3.

**Figure 4 viruses-11-00500-f004:**
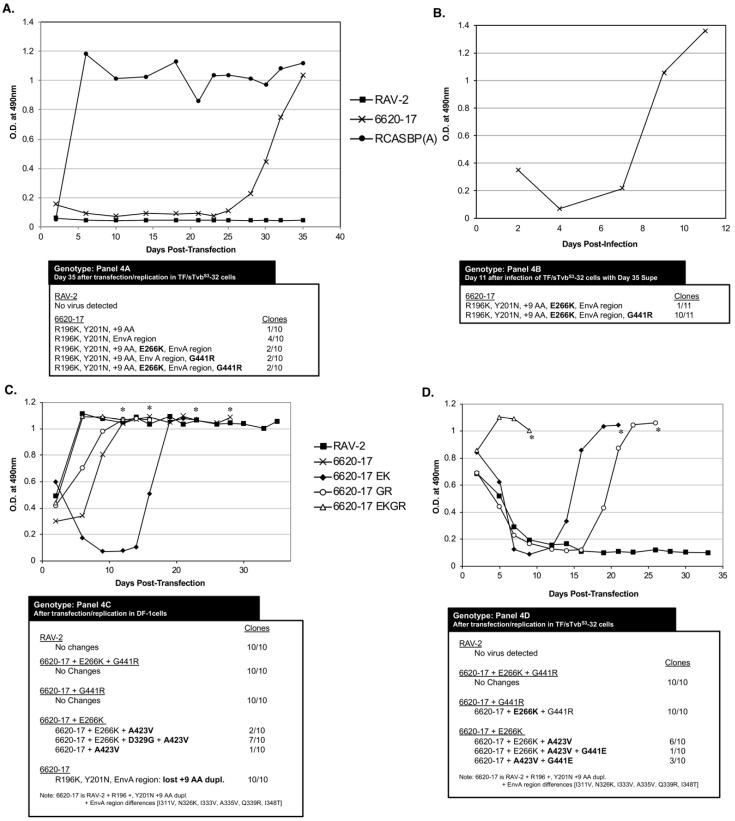
Selection of 6620-17 virus variants resistant to sTvb^S3^-mIgG. Viral growth was monitored by ELISA for the ASLV CA protein. The ASLV *env* nucleotide sequences were determined from clones generated from PCR amplified ASLV *env* sequences from DNA isolated from infected cells. (**A**) Plasmids encoding the ASLVs were transfected in TF/sTvbS3-32 cells expressing high levels of sTvb^S3^-mIgG protein and the cell cultures passaged to allow virus replication. (**B**) The 6620-17 virus pool in the day 35 supernatant was re-passaged in TF/sTvbS3-32 cells. (**C**,**D**) Plasmids encoding the ASLVs were transfected in parental DF-1 cells (**C**) and TF/sTvbS3-32 cells (**D**) and the cell cultures passaged to allow virus replication. The day the cultures began a transient period of ASLV induced cytotoxicity are marked with an asterisk (*).

**Figure 5 viruses-11-00500-f005:**
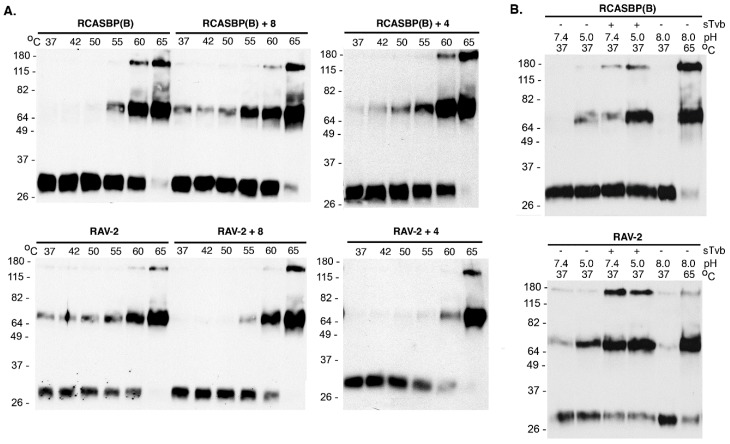
Representative TM glycoprotein oligomerization assays. Shown are examples of the abilities of purified RCASBP(B) and RAV-2 and their mutant Env trimers to form stable, SDS-resistant TM oligomers from infected DF-1 cells. The abilities of Env trimers on ASLV virions to be triggered to form stable TM oligomer confirmations by temperature alone (**A**) and by prior binding to the sTvb^S3^-mIgG receptor and low pH exposure at 37 °C (**B**) are shown. Molecular sizes (in kilodaltons) are given on the left. The separated proteins were analyzed by Western immunoblotting using anti-ASLV(A) or ASLV(B) TM peptide sera followed by HRP-conjugated goat anti-rabbit IgG.

**Figure 6 viruses-11-00500-f006:**
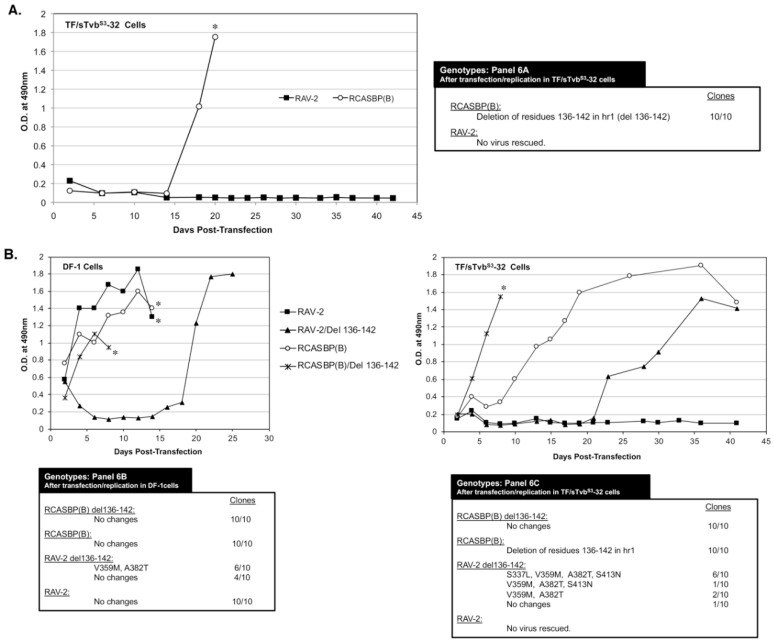
Selection of ASLV(B) viruses resistant to sTvb^S3^-mIgG. Viral growth was monitored by ELISA for the ASLV CA protein. The ASLV *env* nucleotide sequences were determined from clones generated from PCR amplified ASLV *env* sequences from DNA isolated from infected cells. (**A**) Plasmids encoding the ASLVs were transfected in TF/sTvbS3-32 cells expressing high levels of sTvb^S3^-mIgG protein and the cell cultures passaged to allow virus replication. (**B**) Plasmids encoding the ASLVs were transfected in parental DF-1 cells and TF/sTvbS3-32 cells and the cell cultures passaged to allow virus replication. The day the cultures began a transient period of ASLV induced cytotoxicity are marked with an asterisk (*).

**Figure 7 viruses-11-00500-f007:**
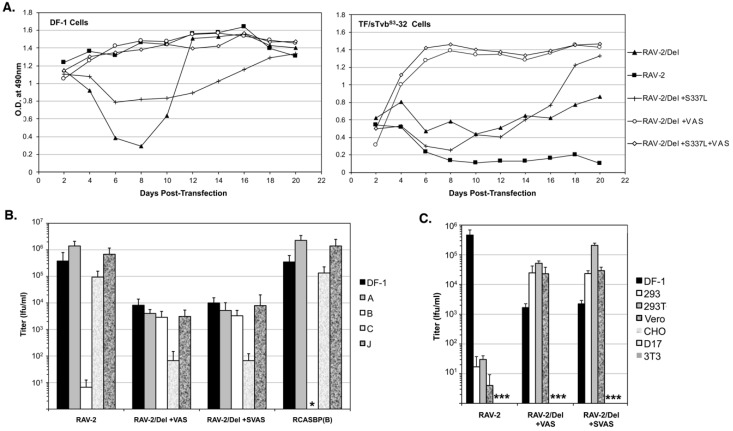
The replicative abilities and receptor usage of RAV-2/Del136-142 mutants. (**A**) Plasmids encoding the ASLVs were transfected in parental DF-1 cells and TF/sTvbS3-32 cells and the cell cultures passaged to allow virus replication. Viral growth was monitored by ELISA for the ASLV CA protein. (**B**,**C**) The abilities of the ASLV parental viruses and mutants to alter receptor usage in avian cells using a receptor interference assay (B) using virus supernatants produced from transfected DF-1 cells (see above) titered on parental DF-1 cells (DF-1) and DF-1 cells previously infected by ASLV(A), ASLV(B), ASLV(C) or ASLV(J). The same viral supernatants were also assayed for their abilities to infect mammalian cells (C) that do not express ASLV receptors. Asterisks (*) denote no detectable titers.

**Figure 8 viruses-11-00500-f008:**
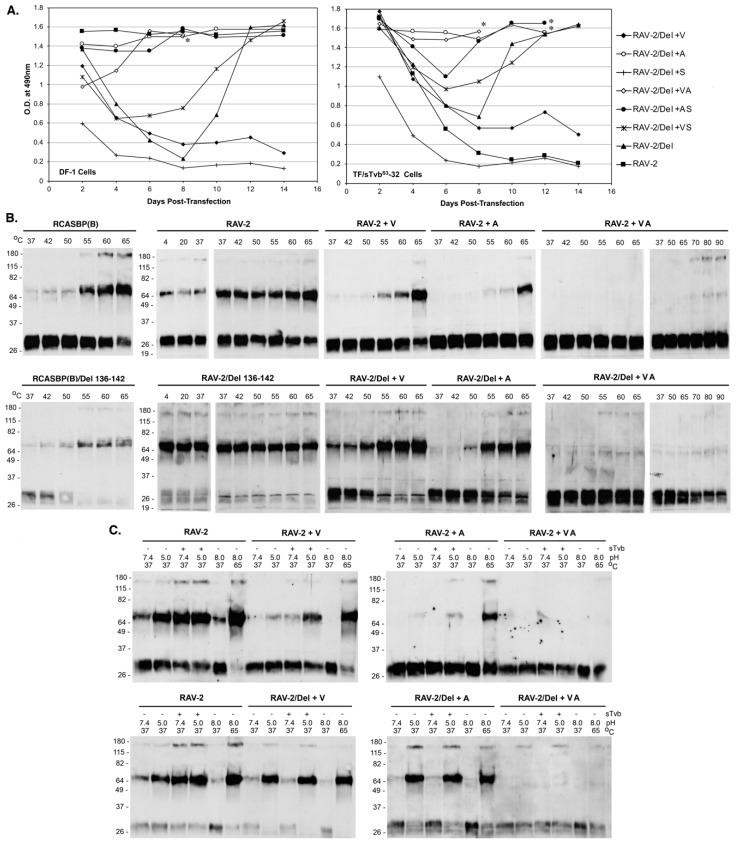
The replicative abilities and representative TM oligomerization assays of RAV-2/Del136-142 mutants. (**A**) Plasmids encoding the ASLVs were transfected in parental DF-1 cells and TF/sTvbS3-32 cells and the cell cultures passaged to allow virus replication. Viral growth was monitored by ELISA for the ASLV CA protein. The day the cultures began a transient period of ASLV induced cytotoxicity are marked with an asterisk (*). (**B**,**C**) The abilities of Env trimers on ASLV virions to be triggered to form stable TM oligomer confirmations by temperature alone (B) and by prior binding to the sTvb^S3^-mIgG receptor and low pH exposure at 37 °C (C) are shown. Molecular sizes (in kilodaltons) are given on the left. The separated proteins were analyzed by Western immunoblotting using anti-ASLV(A) or ASLV(B) TM peptide sera followed by HRP-conjugated goat anti-rabbit IgG.

**Table 1 viruses-11-00500-t001:** Relative resistance of DF-1 cells expressing sTvb^S3^-mIgG to ASLV infection.

Culture	Titer (mean ± SD) ^a^		Average Concentration of sTvb^S3^-mIgG (ng/mL) ^c^
	RCASBP(B)AP [Resistance ^b^]	RCASBP(C)AP [Resistance]	
Uninfected DF-1	(2.9 ± 0.8) × 10^4^	(1.4 ± 0.6) × 10^5^	n.d.
RCASBP(A)	(6.4 ± 3.0) × 10^4^ [0.45]	(2.0 ± 1.2) × 10^5^ [0.70]	n.d.
RCASBP(A)stvb^S3^-mIgG	(4.4 ± 2.3) × 10^2^ [70]	(1.1 ± 0.7) × 10^5^ [1.3]	475 ± 12 (11 nM)
TF/sTvb^S3^-32 cell line	(4.0 ± 2.2) × 10^1^ [725]	(6.5 ± 2.4) × 10^4^ [2.2]	1758 ± 81 (40 nM)

^a^ Serial dilutions of the virions were preabsorbed with supernatant from a confluent culture prior to the assay. ^b^ The resistance of the cells to virus infection was determined by dividing the mean titer obtained on the control uninfected DF-1 cells by the mean titer obtained for each experimental group [shown in brackets]. ^c^ Concentration of sTvb^S3^-mIgG protein in the supernatants as quantitated by ELISA for the mouse IgG tag. The molar concentration of the sTvb^S3^-mIgG protein is given in brackets. The apparent molecular weight of sTvb^S3^-mIgG was calculated from the amino acid sequence and found to be 41,947.

**Table 2 viruses-11-00500-t002:** Chickens expressing sTvb^S3^-mIgG are resistant to ASLV(B) infection but not ASLV(C) infection.

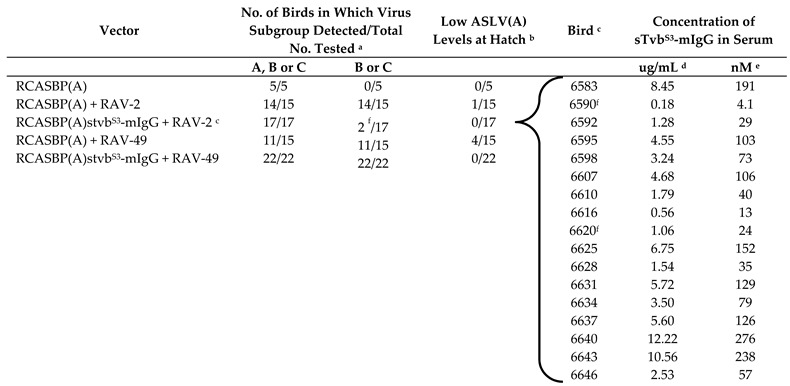

^a^ Serum samples were cultured on DF-1 cells (C/E) that support the replication of ASLV(A), ASLV(B), and ASLV(C); and on line *alv*6 CEFs (C/A) [38] that support ASLV(B) and ASLV(C) but not ASLV(A) replication. ELISA assays detecting ASLV capsid protein (CA) in serum were performed 8 weeks after virus challenge. A positive sample was defined as having an optical density reading of >0.200. ^b^ Birds that had low levels of ASLV CA at hatch (optical densities of 0.23–0.42) did not produce detectable levels of ASLV CA after virus challenge. ^c^ Each bird of this experimental group is listed. ^d^ The concentration of sTvb^S3^-mIgG in the serum of each bird was quantitated by ELISA for the mouse IgG tag. ^e^ The molar concentration of the sTvb^S3^-mIgG protein was determined after calculating the apparent molecular weight of sTvb^S3^-mIgG from the amino acid sequence (41,947). ^f^ The two birds that were positive for ASLV(B) virus. RAV-2 is a subgroup B ASLV isolate. RAV-49 is a subgroup C ASLV isolate.

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
