# Peer review of "Mutations in Both the Surface and Transmembrane Envelope Glycoproteins of the RAV-2 Subgroup B Avian Sarcoma and Leukosis Virus Are Required to Escape the Antiviral Effect of a Secreted Form of the TvbS3 Receptor †"

_viruses, 2019, doi:10.3390/v11060500_

Round 1
Reviewer 1 Report
The manuscript by Yin et al. reports on results from molecular evolution studies of the envelope protein of ASLV subgroup B viruses. The evolutionary pressure on the replicating virus was applied by a soluble form of Tvb. The studies applied both in vivo and in vitro replication and the characterization of the mutants included biochemical studies of envelope proteins. Interestingly, mutations outside the receptor-binding regions of the envelope proteins affected viral fitness, presumably through an influence on conformational transitions as needed for the two-step mechanism of the triggering of membrane fusion characteristic for ASLV.
However, in its present form, the manuscript in not well organized with much focus on detail and lack of clarity in overall conclusions. In particular the Results and Discussion section gives a detailed record of the stepwise selection of mutants which is hard to follow for the reader. Maybe one or more flow diagrams could replace part of the text. Section 4. Conclusions might also be improved for emphasis on the important findings and by the inclusion of more references to other studies.
Moreover, the text needs careful editing at several places:
Examples are:
Line 29; for instead of form
Line 46-47; incorrect sentence
Line 74; evolution and evolve are redundant
Line 103, lack of reference in text
Line 359; a bit of a challenge, not a stringent expression
Line 366; four birds, I count only three
Line 585: that is missing
Line 587; stumbled upon, not a stringent expression
Line 604; to remind, not concise scientific writing
Line 604; since we have yet to…, not concise scientific writing
Altogether, this manuscript needs significant improvement in organization and careful editing.
Author Response
viruses-507103
Reply to Reviewer Report Reviewer 1 (in blue)
However, in its present form, the manuscript in not well organized with much focus on detail and lack of clarity
in overall conclusions. Maybe one or more flow diagrams could replace part of the text.
We have lighted edited the manuscript to improve clarity but we did not radically reorganize the manuscript
since the other 2 Reviewers thought the paper read well.
Section 4. Conclusions might also be improved for emphasis on the important findings and by the inclusion of
more references to other studies.
Section 4: Conclusions has been improved to emphasize the novel conclusions as recommended.
Moreover, the text needs careful editing at several places:
Examples are:
Line 29; for instead of form
Line 46-47; incorrect sentence
Line 74; evolution and evolve are redundant
Line 103, lack of reference in text
Line 359; a bit of a challenge, not a stringent expression
Line 366; four birds, I count only three
Line 585: that is missing
Line 587; stumbled upon, not a stringent expression
Line 604; to remind, not concise scientific writing
Line 604; since we have yet to…, not concise scientific writing
All of these edits have been made as well as many additional corrections as identified in the Word document.
Reviewer 2 Report
Yin et al. studied mutatios arising in Env glycoproteins of subgroup B ASLV due to selected pressure from the soluble receptor. Mutations were found in both SU and TM regions of Env.
Explain sTvb53-mIgG better. Is is the entire receptor? Why is it fused to mIgG/
Line 46-7: fix grammar
Line 255: Not clear why they were concerned about expressing the receptor protein because the virus is toxic.
Line 285: move "by FACS" after "were assayed" in the same sentence.
Author Response
viruses-507103
Reply to Reviewer Report Reviewer 2 (in blue)
Explain sTvbS3-mIgG better. Is this the entire receptor? Why is it fused to mIgG/
We added additional information explaining this was the extracellular domain of the TvbS3 receptor protein
fused to a mouse IgG tag which allowed the use of the extensive anti-mouse IgG reagents to quantitate, purify
and track the immunoadhesin.
Line 46-7: fix grammar
corrected
Line 255: Not clear why they were concerned about expressing the receptor protein because the virus is toxic.
Clarified in the text that over expression of the receptor could have resulted in toxicities in its own right.
Line 285: move "by FACS" after "were assayed" in the same sentence.
corrected
Reviewer 3 Report
The manuscript describes mutations in ASLV-B envelope required to escape the effects of the soluble Tvb receptor inhibition. It further describes in detail the mechanism of action of these mutation and performs biochemical experiments to address this. It shows surprising and novel effects of TM subunit identity on the receptor usage and entry mechanism. The work is highly interesting and very well presented. Although very long, the text reads well and is logically organized and explained.
Few minor points:
1. Line 86 … explain abbreviation SR
2. Line 300 … AP-based titers, how long after infection were they evaluated?
3. Table 2 … in the legend (a), very shortly explain the principle of testing positivity on C/A and C/E cells. Quite unclear how to interpret the numbers now.
4. Line 362 … is there a reference for the resistence of the TF/SUA cells?
5. Line 366 … not clear what four birds are mentioned
6. Line 435 … typo (highlighted in red)
7. Line 448 … possibly a list of mutations in animal 6620 should be mentioned in previous chapter
8. Line 473 … not clear what is the 6620-17-5‘ and 3‘ mutants
9. Line 513 … altgough not necessary, it would be good to show the data of binding
10. Line 523 … perhaps adding short conclusion of this chapter
11. Line 531 … should Fig2 be references instead of Fig1?
12. Line 577 … not clear what is meant with conserved AA (conserved in ASLV subgroups?, but they differ in RCASBP vs RAV2)
13. Fig7 legend … not clear whether the asterisks described in legend (cytotoxicity)applies to the asterisks in 7C.
14. Most figures – the font describing the viruse sis very small, hardly readable.
15. Line 692 … typo, RVA2à RAV2
16. Any speculation why subgroup C would interfere with mutant RAV-2 (Fig 7C)?
17. Discussion Line 761 … Another potential disadvantage of broad tropism should possibly be mentioned – increased cytotoxicity, through accumulation of unintegrated viral DNA (as described in reference Rainey et al)
18. The viral env mutants described (eg lines 710-720) seem to be more stable, because they are less prone to TM oligomer formation at high temperatures. This could possibly be discussed in relation to very similar mutants (by position in ASLV env) described in Lounkova et al, PNAS (PMID: 28607078). These also extend tropism, but seem to be less stable.
Author Response
viruses-507103
Reply to Reviewer Report Reviewer 3 (in blue)
Few minor points:
1. Line 86 … explain abbreviation SR
clarified: subgroup A Scmidt-Ruppin ASLV (SR-A)
2. Line 300 … AP-based titers, how long after infection were they evaluated?
clarified: two days after infection
3. Table 2 … in the legend (a), very shortly explain the principle of testing positivity on C/A and C/E cells. Quite
unclear how to interpret the numbers now.
Clarified both in text and Table legend
4. Line 362 … is there a reference for the resistence of the TF/SUA cells?
added reference
5. Line 366 … not clear what four birds are mentioned
corrected: three birds
6. Line 435 … typo (highlighted in red)
corrected
7. Line 448 … possibly a list of mutations in animal 6620 should be mentioned in previous chapter
The mutations of 6620-17 are listed both in Figure 2 and the sequence in Fig. 3.
8. Line 473 … not clear what is the 6620-17-5‘ and 3‘ mutants
deleted this line
9. Line 513 … altgough not necessary, it would be good to show the data of binding
We did not include the binding data since all of the mutants did not detectably bind the level of sTvbS3-mIgG
used in the assays. A binding figure for wild-type subgroups B and D glycoproteins is included already in Fig.
1C as a reference.
10. Line 523 … perhaps adding short conclusion of this chapter
Conclusions added
11. Line 531 … should Fig2 be references instead of Fig1?
corrected to reference Figure 3
12. Line 577 … not clear what is meant with conserved AA (conserved in ASLV subgroups?, but they differ in
RCASBP vs RAV2)
Clarified to 4 conservative amino acid substitutions versus 4 non-conserved substitutions.
13. Fig7 legend … not clear whether the asterisks described in legend (cytotoxicity)applies to the asterisks in
7C.
Corrected
14. Most figures – the font describing the viruse sis very small, hardly readable.
Enlarged appropriate figure legends
15. Line 692 … typo, RVA2à RAV2
corrected